# Investigating the Inhibition of FTSJ1, a Tryptophan tRNA-Specific 2′-O-Methyltransferase by NV TRIDs, as a Mechanism of Readthrough in Nonsense Mutated CFTR

**DOI:** 10.3390/ijms24119609

**Published:** 2023-06-01

**Authors:** Pietro Salvatore Carollo, Marco Tutone, Giulia Culletta, Ignazio Fiduccia, Federica Corrao, Ivana Pibiri, Aldo Di Leonardo, Maria Grazia Zizzo, Raffaella Melfi, Andrea Pace, Anna Maria Almerico, Laura Lentini

**Affiliations:** Department of Biological, Chemical, and Pharmaceutical Sciences and Technologies, University of Palermo, 90128 Palermo, Italy; pietrosalvatore.carollo@community.unipa.it (P.S.C.); giulia.culletta@unipa.it (G.C.); ignazio.fiduccia@unipa.it (I.F.); federica.corrao01@unipa.it (F.C.); ivana.pibiri@unipa.it (I.P.); aldo.dileonardo@unipa.it (A.D.L.); mariagrazia.zizzo@unipa.it (M.G.Z.); raffaella.melfi@unipa.it (R.M.); andrea.pace@unipa.it (A.P.); annamaria.almerico@unipa.it (A.M.A.)

**Keywords:** FTSJ1, methyltransferase, tRNA, readthrough, stop codon mutation, small molecules, docking, molecular dynamics, MM-GBSA

## Abstract

Cystic Fibrosis (CF) is an autosomal recessive genetic disease caused by mutations in the CFTR gene, coding for the CFTR chloride channel. About 10% of the CFTR gene mutations are “stop” mutations that generate a premature termination codon (PTC), thus synthesizing a truncated CFTR protein. A way to bypass PTC relies on ribosome readthrough, which is the ribosome’s capacity to skip a PTC, thus generating a full-length protein. “TRIDs” are molecules exerting ribosome readthrough; for some, the mechanism of action is still under debate. We investigate a possible mechanism of action (MOA) by which our recently synthesized TRIDs, namely NV848, NV914, and NV930, could exert their readthrough activity by in silico analysis and in vitro studies. Our results suggest a likely inhibition of FTSJ1, a tryptophan tRNA-specific 2′-O-methyltransferase.

## 1. Introduction

Protein synthesis is a crucial phase for any living organism; therefore, any phenomenon affecting any of the steps involved in such a crucial process could cause severe disorders. Protein synthesis goes on until the ribosome meets a stop codon (UGA, UAA, or UAG), thus ending translation and releasing the synthesized polypeptide [1].

Recognition of the stop codon by the translation machinery is essential to terminating translation at the right position and to synthesizing a protein of the correct size. When UGA, UAA, and UAG codons appear earlier in the mRNA sequence, with respect to the normal stop signal due to point mutations in the DNA sequence (nonsense mutations), they are interpreted as a premature termination codon (PTC). Once the ribosome encounters a PTC in its acceptor (A) site, translation will be terminated, thus producing a truncated and nonfunctional polypeptide. Although translation termination must be a very efficient process to ensure the correct protein size, under certain conditions or at a very low rate, a near-cognate tRNA can be recruited to the A site of the ribosome when the latter reaches a stop codon. This event is called stop codon readthrough. Several types of readthrough can occur at stop codons, depending on the presence of regulatory elements or of readthrough-promoting molecules. In the absence of any readthrough molecules, readthrough of any natural stop codon or PTC can occur at a basal level (non-programmed translational readthrough) [2]. The second type of readthrough is called programmed translational readthrough. It targets specific mRNAs [3,4,5], and it is a proteome-expanding mechanism allowing the synthesis of specific protein isoforms. The third type of readthrough is promoted by certain molecules, and it is termed induced stop codon readthrough.

The basal readthrough level varies from one stop codon to another, as shown in various studies [6,7]. The identity of the stop codon influences this level, but other elements acting in a “cis” or “trans” manner can also modulate the efficiency of PTC or natural stop codon readthrough. Such elements can influence all three types of stop codon readthroughs. Cis elements activating stop codon readthrough can be altered by the nucleotide context of the stop codon. Some endogenous trans elements have been identified as proteins or RNAs required either for the readthrough of specific stop codons or for the general readthrough mechanism [2].

Unfortunately, nonsense mutations and PTC are the cause of about 11% of all genetic disorders in humans, such as cystic fibrosis (CF), Duchenne muscular dystrophy (DMD), spinal muscular atrophy, neurofibromatosis, retinitis pigmentosa, lysosomal storage disease, ataxia telangiectasia (AT), Hurler’s syndrome (HS), Rett syndrome, Shwachman–Diamond syndrome, Usher’s syndrome (USH), Hemophilia A and B, Tay–Sachs disease, and several forms of cancer [8].

PTC suppression therapy could help the treatment of patients with nonsense-mediated diseases. In vitro, ex vivo, and in vivo experiments and clinical trials have identified a diverse structural set of nonsense suppressors as candidates for PTC suppression therapy, including aminoglycosides, ataluren (PTC124), ataluren-like molecules, and others [9]. In this context, our research group has more than ten years of experience in this research field [10,11,12,13]. We have identified several compounds endowed with readthrough activity, and three 1,2,4-oxadiazole derivatives (NV848, NV914, and NV930) (Figure 1) are undergoing in vivo studies [14,15].

Even though other nonsense suppressors have been identified in the past years, the elucidation of their molecular mechanism of action remains a crucial point to be addressed with the aim of identifying more active compounds.

In a recent manuscript, in vitro kinetic assays on eukaryotic translation suggested different mechanisms for different nonsense suppressors. Aminoglycosides via binding a single tight site on the ribosome and ataluren-like compounds via weaker multisite bindings induce a slower change in the protein synthesis apparatus that permits readthrough [16].

In another recent study, an extract of the mushroom *Lepista inversa* showed a high-efficiency correction of UGA and UAA nonsense mutations [17]. One active constituent of this extract is a 2,6-diamino purine (DAP), which has been shown to increase p53 levels in Calu-6 cancer cells. DAP interferes with the activity of a tRNA-specific 2′-O-methyltransferase (FTSJ1) responsible for cytosine 34 modification in tRNA-Trp.

Starting from this recent experimental evidence on small molecules, in this manuscript, we tried to elucidate if the molecular mechanism of actions of our readthrough compounds NV848, NV914, and NV930 could be attributable to the inhibition of the tRNA-specific 2′-O-methyltransferase (FTSJ1). An investigation by combined computational approaches followed by Luciferase assay evaluation in HeLa cells upon transfecting an FTSJ1 cDNA harboring plasmid aided the purpose of this study. In addition, because our NV848, NV914, and NV930 compounds have been shown to rescue both protein levels and functionality of the CFTR (Cystic Fibrosis Transmembrane Conductance Regulator) chloride channel [14], mutated in CF, we also tested for CFTR protein cellular localization in IB3.1 cystic fibrosis cells [18] upon FTSJ1 transfection.

## 2. Results and Discussion

### 2.1. Homology Modeling

Recent studies demonstrated that small molecules could interfere with proteins or RNAs required either for the readthrough of specific stop codons or for the general readthrough mechanism [2]. In this context, a small molecule such as 2,6-diamino purine (DAP) showed to promote the readthrough of premature stop codons of the p53 transcript by inhibiting a specific methyltransferase known as FTSJ1. In the attempt to elucidate the mechanism of action of our readthrough compounds NV848, NV914, and NV930, we performed a series of computational studies with increasing accuracy: blind docking (BD), extra precision docking, and unbiased molecular dynamics (MD) followed by MM-GBSA (molecular mechanics-generalized Born surface area) free energy calculation followed by in vitro analysis. We began computational studies exploring the Protein DataBank (PDB) in the search for the experimental structure of the methyltransferase object of this study. Two structures were retrieved both related to the X-ray structures of the yeast tRNA methyltransferase (PDB ID: 6JP6 and 6JPL), the last one in complex with S-adenosyl methionine (SAM). The human homolog protein FTSJ1 was modeled in the SWISS-MODEL workspace (swissmodel.expasy.org/workspace) using the PDB ID: 6JP6 as a template. The structures have been optimized by completing and refining the missing loops and residues and optimizing amide groups of asparagine (Asn) and glutamine (Gln), and the imidazole ring in histidine (His), and predicting protonation states of histidine, His, aspartic acid (Asp), and glutamic acid (Glu) and tautomeric states of histidine.

### 2.2. Blind Docking and Semi-Flexible Docking Analysis

Blind docking analysis was performed using Achilles Blind docking server (https://bio-hpc.ucam.edu/achilles/entry) with the homology model of the human FTSJ1 previously obtained against DAP, NV848, NV914, NV930, PTC124, and SAM. The binding position is located around Lys156, representing the active site as a proton acceptor, and Gly53, Trp55, Asp75, Asp91, and Asp116 as residues that constitute the binding site. Blind docking performs an exhaustive series of docking calculations across the whole protein surface to find the spots with the best binding affinities. After the affinities are calculated, the tool clusters the results according to the spatial overlapping of the resulting poses. For each cluster, the pose with the best affinity is taken as the representation of this cluster. Interestingly, the best pose calculated in terms of binding energy (kcal/mol) for each compound falls in the binding site, as previously reported. In Figure 2, the best cluster poses of DAP are reported as representative evidence of the blind docking procedure.

The blind docking analysis result was very useful in identifying the most probable binding site, but in order to have more accuracy in terms of the pose orientation, the extra precision semi-flexible docking was performed centering the docking box on the 3D coordinates of SAM as reported in the complex PDB ID: 6JPL. In semi-flexible docking, the ligands are free to change their conformational structure around the six rotational degrees of freedom, while the target is considered static. As validation of the extra precision scoring function, SAM was redocked, and the RMSD was calculated, showing a value <0.3 Å (Figure 3).

Successively, DAP, NV848, NV914, NV930, and PTC124 were docked in the binding pocket of FTSJ1 to define the most likely pose and the interactions with the target protein. The docking calculation was performed in Glide extra precision mode to obtain the best accuracy. For each compound, we stored up to 10 poses with the best docking score, and after a visual inspection, the poses with the best scores were selected to perform molecular dynamics simulations (Figure 4).

In two cases, for NV848 and NV930, we also selected the second best pose due to the almost equal score but with a quite different pose in the binding pocket, as reported in Figure 5.

### 2.3. Molecular Dynamics and Free Energy Analysis

The running of molecular dynamics simulations of target–ligand complexes over time is considered the most accurate approach in computer-aided drug design. Here, an unbiased molecular dynamics simulation was performed to investigate the conformational stability and the time-dependent binding capability of DAP, as a known FTSJ1 inhibitor, NV848, NV914, NV930, and PTC124 in the active site of FTSJ1, as potential inhibitors. Moreover, we performed MD simulations on the complex FTSJ1/SAM as a control. We explored if the protein target undergoes conformational alterations interacting with the compounds. Three replicas of 100 ns each of MD simulations were carried out starting from the docking poses previously obtained: FTSJ1/DAP, FTSJ1/NV848, FTSJ1/NV848_pose2, FTSJ1/NV914, FTSJ1/NV930, FTSJ1/NV930_pose2, and FTSJ1/PTC124. While three replicas of 100 ns each of MD simulations were carried out for SAM starting from the co-crystallized pose, each replica of 100 ns was performed by randomizing the initial velocities and seeding with the aim of sampling the conformational space and obtain a total of 300 ns of simulation for each system. Various analyses, such as root-mean-square deviation (RMSD), root-mean-square fluctuation (RMSF), and determination of the types of protein–ligand contacts, were performed to obtain a more detailed analysis of the target–ligand complexes.

The RMSD has been selected as a criterion to evaluate the dynamic stability of ligand-bound systems. All protein frames are first aligned on the reference frame backbone, and then the RMSD is calculated based on the atom selection, in these cases on the Cα. For these complexes, the RMSD values of the protein’s Cα atoms and ligand are reported in Figure 6.

As a reference system, we first performed the simulations of FTSJ1/SAM. This system reached the equilibrium in the first steps of the simulations and fluctuated around the average of 3 Å. Similar behavior is evidenced in all the complexes with DAP, NV848, NV914, NV930, and PTC124, where the average value of RMSD of the proteins Cα atoms is comprised between 2.5 Å and 3.9 Å. Once defined that the simulations converged, it would be useful to evaluate the ligand RMSD. The ligand RMSD indicates how stable the ligand is with respect to the protein and its binding pocket. This plot evidence the RMSD of a ligand when the protein–ligand complex is first aligned on the protein backbone of the reference, and then the RMSD of the ligand-heavy atoms is measured. If the values observed are significantly larger than the RMSD of the protein, then it is likely that the ligand has diffused away from its initial binding site (Figure 7).

Starting from the reference system FTSJ1/SAM, the native ligand maintains a stable value of RMSD and confirms the reliability of the simulations. The same behavior is shown by DAP. When the ligand RMSD is analyzed for NV848, it could be noted that a dual behavior from the two different starting poses was obtained by the docking. In pose 1, just replica 2 maintains a stable RMSD value for the entire simulation, while in replica 1, the ligand flies away from the binding pocket after about 83 ns of the simulation. In replica 3, this event is observed after 23 ns. On the opposite replicas 1 and 3 of NV848, pose 2 showed stable ligand RMSD values around 3 Å. In replica 2 of NV848 pose 2, there is a shift around 44 ns of the simulation of the RMSD values, but in this case, the ligand remains in the binding pocket, just fluctuating in a different orientation from the beginning. The three replicas of NV914 showed a similar behavior of the ligand with stable RMSD values with an average of 7–8 Å. The visual inspection of the three simulations shows that the ligand is stable in the interactions with the pocket, and the RMSD fluctuations are ascribed to the movements of the carboxyamideperfluoroaryl ring that points outside the pocket. As previously defined, two poses were selected to perform the MD simulation of NV930. The first pose in all three replicas flies out from the binding, as shown in Figure 7f. Only in replica 1 the ligand maintains its position in the binding pocket for about 28 ns. The NV930 pose 2 is stable in replica 3. In replicas 1 and 2, even though the ligand steps away from the starting pose, it remains permanently in the binding pocket establishing new interactions. In the end, the three replicas of PTC124 showed stable RMSD values. Replicas 2 and 3 show an average RMSD value of 2.2 Å.

The main chain average RMSF of the complexes has been calculated for the entire 100 ns of each replica to examine the structural flexibility effect of the compounds upon the FTSJ1 binding pocket. The residue-wise fluctuation of the complexes was plotted and presented in Figure 8. The plot has been coupled according to the different ligands bound to compare the chain behavior due to different ligands. It is worth noting that the RMSF values reported in the plot are the average of the three replicas. The analysis of the results of the residues’ mobility was focused on the residues that constitute the binding site (Gly53, Trp55, Asp75, Asp91, Asp116, and Lys156). The RMSF values of these residues are under 2 Å: Gly53 1.1 Å, Trp55 1.9 Å, Asp75 1.2 Å, Asp91 0.9 Å, and Lys156 1.0 Å. Just Asp116 showed an average RMSF value of 2.7 Å but slightly higher in the PTC124 and NV848 replicas. The outcome of this analysis revealed that the binding of these small molecules did not influence in a concrete way the binding pocket.

Estimation of protein interactions provides a measure of interaction power between the ligands and the target protein. Protein interactions with the ligand can be monitored throughout the simulations. In this section, the interactions of the ligands with the target protein will be evaluated, starting from the poses selected with the docking analysis and the interactions observed during the MD simulations. All the interactions observed in the docking analysis and in the MD simulations are reported in Table 1. It is worth noting that the interactions observed during the MD simulations are reported for each simulation. Taking into consideration that NV848 and NV930 show a dynamic behavior that is not stable, only the interactions observed in pose 2 are considered for the comparative analysis with the docking interactions.

Moreover, in the comparative analysis, the residues involved in the interactions should be stressed much more than the kind of interactions observed.

The known FTSJ1 inhibitor DAP interacts with Trp55 by a pi–pi interaction, with Lys156 by a pi–cation interaction, and with Asp47 and Asp116 by vdW interactions in the docking analysis. In the three replicas of the MD simulation, the interactions with Trp55, Asp47, and Asp116 are conserved in terms of hydrophobic and H-bonds interactions. New hydrophobic interaction with Leu48 is observed in all three replicas and H-bonds with Gly53 and Ser54 in replicas 2 and 3.

NV848 shows the same interactions with Trp55 and Asp116, as observed with DAP in the docking analysis. In addition, an H-bond with Trp55 and two vdW interactions with Gly53 and Ser54 are registered. In the three simulations along the 100 ns, the interactions Gly53, Ser54, and Asp116 are conserved with respect to the docking analysis in terms of H-bonds and a water bridge with Asp116. Moreover, the interaction with Trp55 is conserved in replicas 1 and 3, and in replica, there is the involvement of Lys156 in a hydrophobic interaction with the ligand. Other interactions with Lys28, Asp47, Leu48, and Cys115 are observed and reported along the time.

NV914 is the compound that shows a slight binding pattern with respect to docking and MD replicas. Indeed, in the docking analysis, it interacts with Cys49, Asp75, Asp116, Ala118, Leu135, and Lys156 just by vdW interactions and one H-bond with Ala118. In the three replicas, it was already underlined that the ligand has RMSD values comprised between 6–9 Å, and these values are reflected in different residues, which are involved in the binding pattern of the compound. The interactions with Cys49 and Ala118 are conserved, and new H-bonds, hydrophobic interactions, and water bridges are observed with Leu76, Ile92, and Asp120.

NV930 established an H-bond with Ala118 and several vdW interactions with Cys49, Gly53, Trp55, Asp75, Asp116, Ala118, Leu135, and Lys156 in the docking analysis. During the MD simulations, the starting pose of NV930 shows higher RMSD values but retains some interactions with previously observed residues of Cys49, Asp75, Ala118, and Leu135, establishing new interactions of Ile92, Leu76, Ala138, Ala139, and Tyr218.

PTC124 shows a pi–cation stacking with Lys156, an H-bond with Arg24, three salt bridges with Arg24, Lys28, and Arg186, and more vdW interactions with Ser25, Cys49, Pro52, Gly53, and Trp55 in the docking analysis. The three replicas retain the major part of the interactions retrieved in the docking analysis, above all in replica 2 and replica 3

To understand the biophysical basis of molecular recognition of the compounds object of this study with FTSJ1, a molecular mechanics-generalized Born surface area (MM-GBSA) approach was used. As a comparative analysis, we performed the binding free energy analysis for all the replicas performed, including the SAM/FTSJ1 replicas. This analysis provides as an outcome the ΔG values as the mean of the simulations snapshot considering the contribution of the water as implicit. The conformational entropy change TΔS was not calculated to reduce the computational time. The ΔG values of all the replicas are reported in Figure 9 together with the mean ΔG values obtained from the three replicas. The ΔG values of the complexes FTSJ1/SAM is reported as a reference of the energetic scale of the binding. Indeed, a natural ligand has a higher binding energy compared with small molecules and inhibitors. The binding energy analysis shows that DAP, PTC124, and NV914 have comparable binding energy, slightly higher than NV848 and NV930. Considering that DAP is a recognized inhibitor of FTSJ1, the docking outcomes, together with the MD replicas and binding energies, suggest that the NV compounds and PTC124 could be potential inhibitors of this target protein.

### 2.4. In Vitro Analysis

We had previously demonstrated the readthrough activity exerted by NV848, NV914, and NV930 via a FLuc (Firefly luciferase) cell-based assay [14,19,20]. There have been some speculations about the molecular mechanisms of readthrough induction upon TRIDs (translational readthrough-inducing drugs), and one accredited hypothesis considers the interaction between these molecules with the mRNA [8]. Following our in silico analyses, we wanted to understand by in vitro tests if our NV848, NV914, and NV930 could induce readthrough by FTSJ1 activity inhibition, as well as DAP and PTC124. To this aim, HeLa cells were transiently co-transfected with a pFLuc190UGA plasmid [19,21] plus a plasmid harboring the FTSJ1 cDNA (a kind gift from Dr. Lejeune) [17]. As control of the transfection, HeLa cells were also transfected with the only pFLuc190UGA (Fluc Opal) plasmid or alternatively with a pFLucWT vector.

Then, 24 h post-transfection, HeLa cells were transfected with only the pFLuc190UGA plasmid, and the cells co-transfected with both the pFLuc190UGA plasmid and the FTSJ1 plasmid were treated with the three readthrough compounds NV848, NV914, or NV930 at 6 µM, 12 µM, or 24 µM, respectively. Twenty-four h later, cells were then assayed for luminescence. As reported in Figure 10, in pFLuc190UGA-transfected HeLa cells (Fluc Opal only, black histograms) treated with either NV848, NV914, or NV930, there is an increased level of luminescence, as expected, compared with the untreated cells (UNT). On the contrary, HeLa cells co-transfect with both the pFLuc190UGA plasmid and the FTSJ1 plasmid (2 µg FTSJ1, dark blue histograms) shows reduced luminescence levels for all the molecules and concentrations tested compared with the only pFLuc190UGA plasmid transfection. This finding suggests a possible inhibitory role exerted by NV848, NV914, and NV930 on FTSJ1, whose overexpression counteracts NV compounds’ readthrough activity.

We also ran a luciferase assay to test if PTC124 readthrough activity could be influenced by the overexpression of FTSJ1. In line with the results of our new TRIDs, overexpression of FTSJ1 decreases the readthrough exerted by PTC124 for all the concentrations tested (Figure 11). Thus, this result strengthens our hypothesis whereby our NV848, NV914, and NV930 molecules could likely exert stop codon readthrough by FTSJ1 inhibition.

It is important to stress the fact that our goal is to find new TRIDs capable of restoring a full-length and functional CFTR protein. Thus, in order to ascertain if NV848, NV914, and NV930 could exert readthrough on CFTR mRNA by inhibiting FTSJ1 activity, IB3.1 cells were used as a cystic fibrosis cell model [18]. IB3.1 cells are characterized by a UGA premature termination codon instead of a UGG tryptophan codon at aminoacidic position 1282 of the CFTR protein sequence, which thus results in a truncated CFTR protein (W1282X mutation). In addition, IB3.1 cells also harbor an F508del mutation, which can result in the residual presence of CFTR at both mRNA and protein levels [11,22,23].

Because we had previously demonstrated NV848, NV914, and NV930’s readthrough activity on a rat cell line upon transfection of a vector harboring the CFTR cDNA with the W1282X mutation [14], we wanted to investigate if the same molecules could have the same readthrough effect on a human cell line of cystic fibrosis, through the inhibition of FTSJ1 activity.

To this aim, IB3.1 cells were plated onto glass coverslips and assayed by immunofluorescence after 24 h treatment with one of the above NV compounds at 12 µM, a concentration we had previously demonstrated to exert ribosome readthrough for CFTR [14]. As positive controls of readthrough induction, either G418 (300 µg/mL) or PTC124 (12 µM) were used [19,22]. As reported in Figure 12a, there is a visible rescue of the fluorescence relative to CFTR protein expression in IB3.1 cells treated with either NV848, NV914, or NV930, compared with the untreated cells (UNT). In addition, the fluorescence intensities in NV848-, NV914-, and NV930-treated cells (quantification in Figure 13a) outnumber the fluorescence intensities in G418- and PTC124-treated cells (fluorescence images in Figure 12b and quantification in Figure 13a), suggesting a higher readthrough induction exerted by NV compounds.

Surprisingly however, upon FTSJ1 plasmid transfection (fluorescence images in Figure 12c and quantification in Figure 13b), CFTR fluorescence intensities decrease in all NV848-, NV914-, and NV930-treated cells (fluorescence images in Figure 12c and quantification in Figure 13c) compared with IB3.1 cells treated only with NV molecules in the absence of FTSJ1 expression (fluorescence images in Figure 12a and quantification in Figure 13c).

Altogether, our results suggest that NV molecules could specifically exert readthrough on UGA PTCs by inhibiting FTSJ1 methyltransferase activity. In fact, upon increased FTSJ1 expression, NV compounds’ readthrough activity decreases.

## 3. Materials and Methods

### 3.1. Homology Modeling and Protein Preparation

The putative tRNA (cytidine(32)/guanosine(34)-2′-O)-methyltransferase—UniProt Q9UET6 (TRM7_HUMAN) Homo sapiens—was modeled in the SWISS-MODEL workspace (swissmodel.expasy.org/workspace). The X-ray structure of yeast tRNA methyltransferase complex of Trm7 and Trm734 essential for 2′-O-methylation at the first position of anticodon in specific tRNAs (PDB ID:6JP6) was used as a template of the human tRNA (cytidine(32)/guanosine(34)-2′-O)-methyltransferase FTSJ1 (50.0% sequence identity). This one was the best available experimental structure at the time of the study. The model obtained has a Qualitative Model Energy ANalysis (QMEAN) score of 0.75. The model was refined using the protein preparation wizard tool of Maestro Suite Software [24]. This tool allowed protein structure optimization, including missing loops, side chains, and hydrogens, optimization of the protonation state in a pH range of 7.0 ± 2.0, and analysis of atomic clashes. The protein was refined using restrained minimization with OPLS2005 as a force field. This model was used for further blind docking analysis.

### 3.2. Blind Docking and Semi-Flexible Docking

Blind docking was performed to analyze and locate the most likely protein–ligand interactions of DAP and NV compounds with the methyltransferase FTSJ1. In the blind docking approach, docking is applied to various locations covering the whole protein surface [25]. These blind docking simulations have been calculated with the help of two different docking programs: Autodock Vina and Lead Finder [26,27]. Both these programs use force-field-based scoring functions, including specific intramolecular and intermolecular values contributing to the overall potential, along with genetic algorithms finding the global minima. The compounds were submitted to a semi-flexible docking study using Glide v9.0 [28] in extra precision (XP) with the OPLS2005 force field. The grid box was built considering SAM as the centroid of the grid. The study was performed using no constraints. The van der Waals radii were set at 0.8, and the partial cutoff was 0.15 with flexible ligand sampling. Bias sampling torsion penalization for amides with non-planar conformation and Epik state penalties were added to the docking score.

### 3.3. Molecular Dynamics and MM-GBSA Calculations

Three MD simulation replicas of 100 ns each were carried out using a Desmond 6.5 [29] using the OPLS4 force field for each complex FTSJ1/DAP, FTSJ1/SAM, and FTSJ1/NV848, NV914, and NV930. The system setup and the simulation options are the same as reported in previous manuscripts [30,31,32,33]. Initial velocities were determined with random seeds. The MM-GBSA approach employs molecular mechanics, the generalized Born model, and the solvent accessibility method to determine free energies from structural information circumventing the computational complexity of free energy simulations wherein the net free energy is treated as a sum of a comprehensive set of individual energy components, each with a physical [34]. We applied this method to the snapshots extracted from the 100 ns production MD trajectories. Protein–ligand binding free energy using MM-GBSA was calculated as the difference between the energy of the bound complex and the energy of the unbound protein and ligand. In this work, MM-GBSA calculations were also achieved in Prime software [35]. The entropy term—TΔS—was not calculated to reduce computational time. The VSGB solvation model was chosen with the minimized sampling method.

### 3.4. Cell Culture and NVs Resuspension

HeLa and IB3.1 cells were cultured in Dulbecco’s Modified Eagle Medium (DMEM) (Corning, Corning, NY, USA) supplemented with 10% Fetal Bovine Serum (FBS) (Corning) at 37 °C with 5%CO_2_. No antibiotics were used. Lyophilized NV848, NV914, and NV930 were weighed and resuspended in 100% Dimethyl Sulfoxide (DMSO) (Merck, Rahway, NJ, USA) solution to reach a stock solution of 100 µM. Then, an initial concentration of 1 µM for each NV was prepared by diluting the stock solution in Dulbecco’s Phosphate Buffered Saline 1X (DPBS 1X) (Corning). The diluted solutions were then used to treat cells.

### 3.5. Luminescence Assay

A total of 2.5 × 10^5^ HeLa cells were plated onto each well of the 6 MW plate the day before transfection. A total of 1 µg of pFLuc190UGA plasmid [19,21] was transfected alone or co-transfected with 2 µg of FTSJ1 plasmid [17] by using Lipofectamine 2000 reagent (Invitrogen, Waltham, MA, USA), according to the provider’s instructions. A total of 1 µg of pFLuc-WT plasmid was transfected alone as a positive control. Twenty-four hours after transfection or co-transfection, HeLa cells were left untreated or treated for 24 h with 6 µm, 12 µm, or 24 µm for each NV. The next day, after two washes in DPBS 1X, cells were incubated for 5 min with 400 µL/well of the detection mix-Steady-Glo luciferase reagent (Promega, Madison, WI, USA). A total of 200 µL of cell suspension was then plated in duplicate onto a 96-well plate. Luciferase activity was then acquired with a luminometer.

### 3.6. Immunofluorescence Microscopy

A total of 10^5^ IB3.1 cells were seeded onto glass coverslips in a 12 MW plate the day before transfection. Next, 2 µg of FTSJ1 plasmid was transfected by using Lipofectamine 2000 reagent (Invitrogen), according to the provider’s instructions. Twenty-four hours after transfection, IB3.1 cells were treated (or left untreated) for 24 h with 12 µm for each NV. G418 (300 µg/mL) or PTC 124 (12 µM) were used as positive controls of readthrough induction [22]. The day after treatment, cells were briefly washed in DPBS 1X and then fixed with 4% Paraformaldehyde (PFA) (Thermo Scientific, Waltham, MA, USA) at room temperature for 15 min. After a wash in DPBS 1X, cells were incubated for 15 min in a solution of Glycin 1 mM (Merck), and then cell membrane and Golgi were stained by using the wheat agglutinin germ (WGA) conjugated with Alexa 594 (Life Technologies, 1:1000, Carlsbad, CA, USA) for 10 min. Cells were then permeabilized with 0.1% Triton X-100 (Merck) for 10 min, washed in DPBS 1X, and blocked with 5% FBS solution for 1 h at room temperature. Cells were incubated with a mouse monoclonal antibody (ab570 1:400) overnight at 4 °C followed by a goat-polyclonal to mouse antibody Alexa-Fluor-488 (Abcam,1:500, Cambridge, UK) for 1 h at room temperature to detect CFTR protein. Cells were incubated with a rabbit polyclonal antibody (Cusabio, 1:100, Houston, TX, USA) overnight at 4 °C followed by a goat-polyclonal to rabbit antibody Alexa-Fluor-647 (Invitrogen, 1:1000) to detect FTSJ1. Nuclei were counterstained by using a solution of 4′,6-Diamidino-2-Phenylindole, Dihydrochloride (DAPI) plus Antifade (Thermo Scientific). Cells were observed under a Zeiss Axioskop microscope (Oberkochen, Baden-Württemberg, Germany) equipped for fluorescence. 

### 3.7. Image Analysis of Immunofluorescence Images

Images were opened by using Fiji software [36], then the channels corresponding to CFTR or FTSJ1 were background subtracted, saved, and single ROIs (Region Of Interest) of the entire cells were manually drawn on the CFTR channel. Mean gray values were used to report fluorescence intensities for every single cell analyzed.

## 4. Conclusions

In this work, we faced the hypothesis of a new putative target for our in-house TRIDs, NV848, NV914, and NV930. We performed a virtual and in vitro study of the interaction between NV molecules and FTSJ1 to test a possible inhibition of this tryptophan tRNA-specific 2′-O-methyltransferase as a potential readthrough MOA. The study has been performed by comparison with DAP and SAM as positive controls for the virtual inhibition study and PTC124 and G418 as positive controls for the readthrough activity.

Despite the readthrough activity demonstrated by all three NV molecules, we can suggest that NV914 might exert readthrough by directly inhibiting the activity of FTSJ1, whereas NV930 could slightly inhibit and NV848 could lower FTSJ1 activity. This can be easily retrieved by looking at the luciferase assay as well as at the immunofluorescence analysis, where NV914 exerts less readthrough in the presence of the overexpressed FTSJ1 compared with both NV848 and NV930. In addition, our results on IB3.1 cells reinforce what has been recently reported [37] concerning patient-derived intestinal organoids (PDIOs) harboring a W1282X mutation that displayed increased swelling activity following DAP treatment.

In conclusion, we demonstrate that FTSJ1 inhibition is a plausible readthrough MOA for NV914 and NV930, as well as for DAP, while, concerning NV848, we must say that, although a certain inhibition effect is observable, this target could be one of the possible, but not the only one way of readthrough action. For instance, as demonstrated for other PTC124 derivatives by our group in the past [10,22], NV848 readthrough activity could be the result of the interaction of this molecule with the PTC in the mRNA sequence, which, thus, would favor either the insertion of a near cognate amino acid or the skipping of the nonsense mutation by the ribosome without the addition of any amino acid [8]. Other possibilities could be the suppression of the NMD pathway as well as the interaction with the translation release factors [8]. However, more experiments are needed to exactly unravel the precise mechanism(s) of action not only for NV848 but also for all our new TRIDs in general.

## Figures and Tables

**Figure 1 ijms-24-09609-f001:**
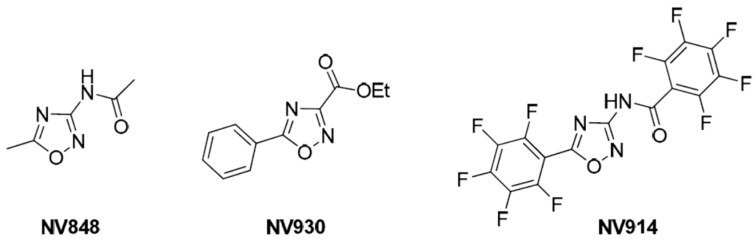
NV848, NV930, and NV914 molecular structures. Image from [14].

**Figure 2 ijms-24-09609-f002:**
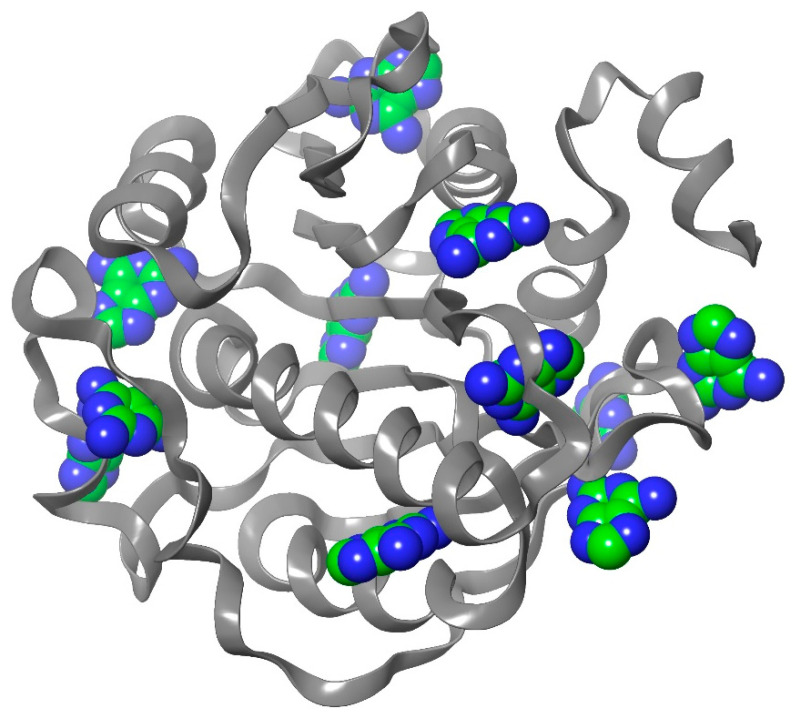
Blind docking of DAP, where carbons are depicted in green, whereas nitrogens are in dark blue.

**Figure 3 ijms-24-09609-f003:**
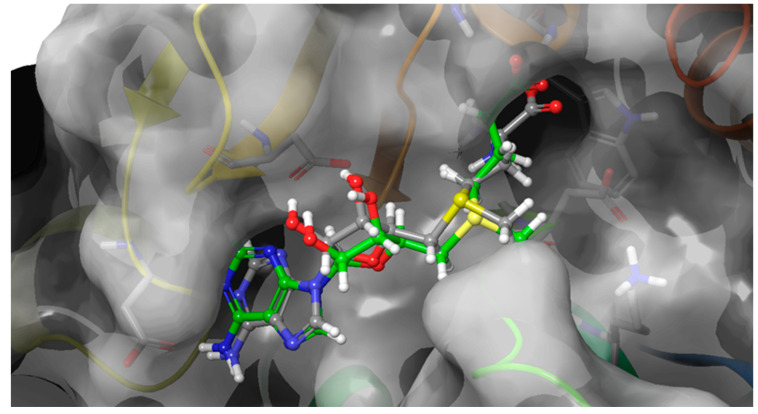
Redocking of SAM in order to validate the Glide XP scoring function. Co-crystallized SAM is reported in green carbons, and redocked SAM is reported in grey carbons.

**Figure 4 ijms-24-09609-f004:**
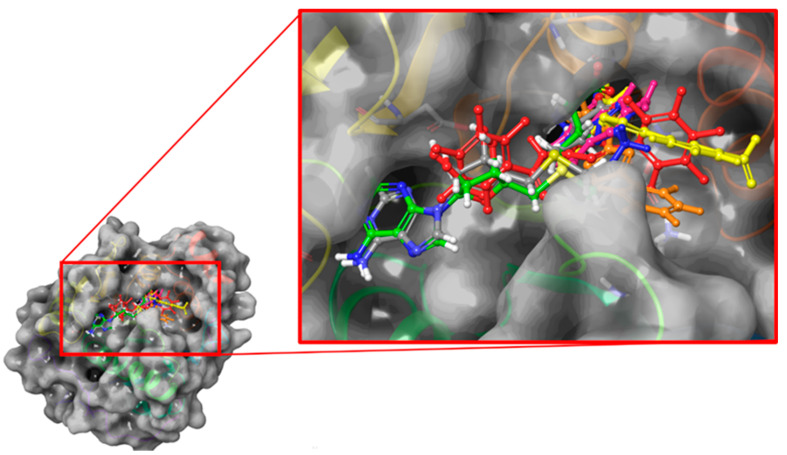
The best docking poses of the molecules object of the study. DAP is reported in purple, NV848 in blue, NV914 in red, PTC124 in yellow, and NV930 in orange. Co-crystallized SAM is reported in green carbons, and redocked SAM is reported in grey carbons.

**Figure 5 ijms-24-09609-f005:**
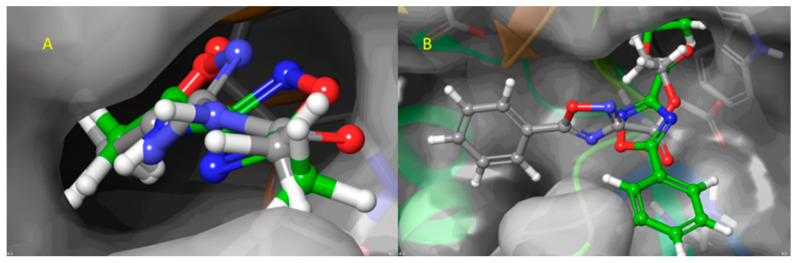
(**A**) Selected poses of NV848 and (**B**) selected poses of NV930 to perform molecular dynamics simulations.

**Figure 6 ijms-24-09609-f006:**
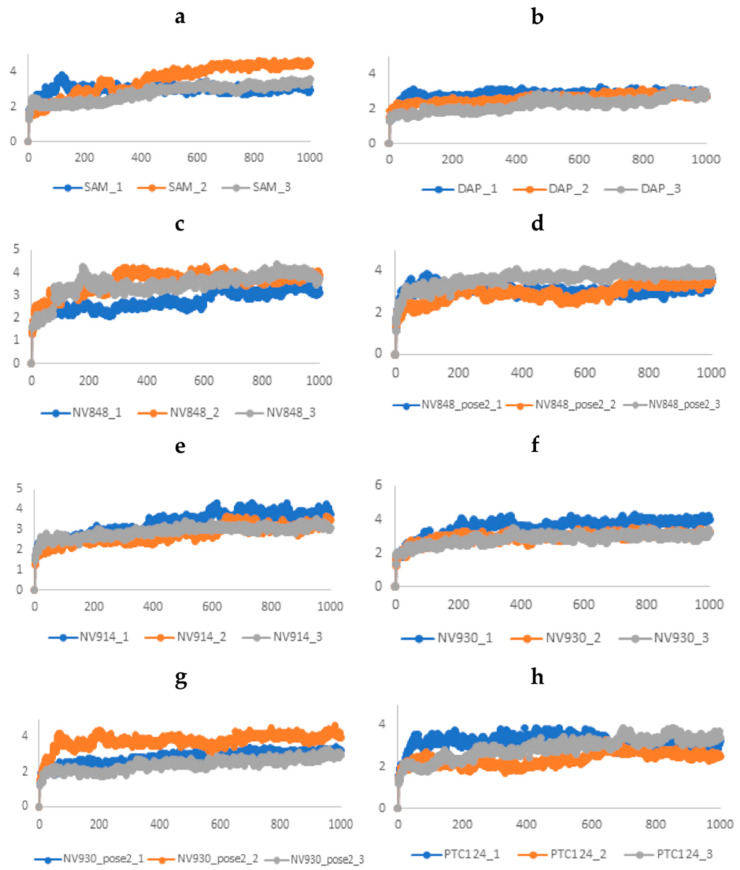
Time evolution of the RMSD (Å) values of Cα atoms of FTSJ1 in complex with (**a**) SAM, (**b**) DAP, (**c**) NV848, (**d**) NV848_pose2, (**e**) NV914, (**f**) NV930, (**g**) NV930_pose2, and (**h**) PTC124. One snapshot corresponds to 0.1 ns.

**Figure 7 ijms-24-09609-f007:**
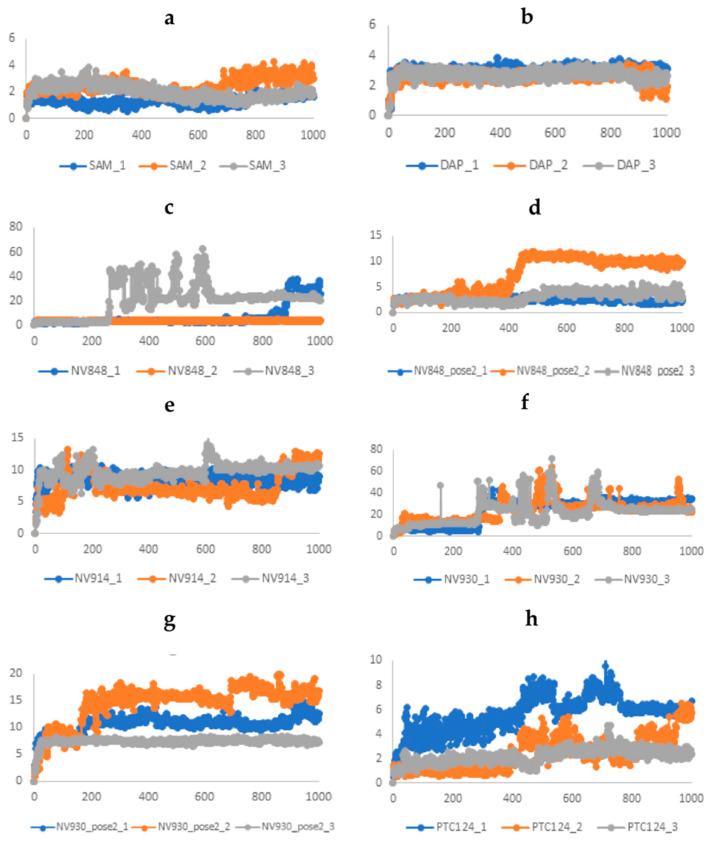
Time evolution of the ligand RMSD (Å) values: (**a**) SAM; (**b**) DAP; (**c**) NV848; (**d**) NV848_pose2; (**e**) NV914; (**f**) NV930; (**g**) NV930_pose2; and (**h**) PTC124. One snapshot corresponds to 0.1 ns.

**Figure 8 ijms-24-09609-f008:**
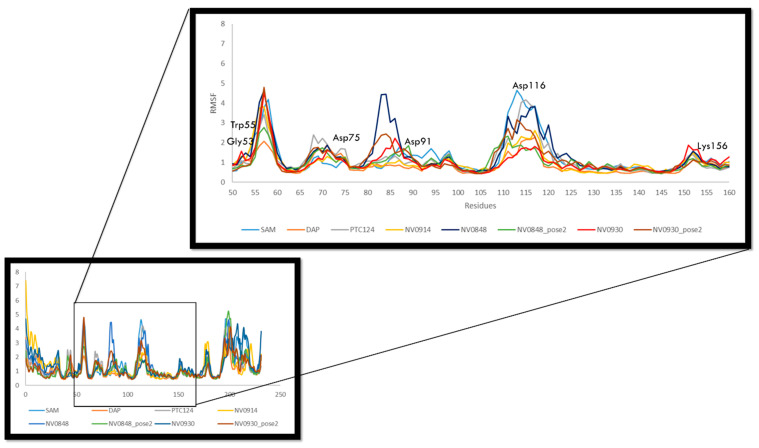
The comparative RMSF values of the backbone atoms (Å). Each curve reports the average value of three replica simulations.

**Figure 9 ijms-24-09609-f009:**
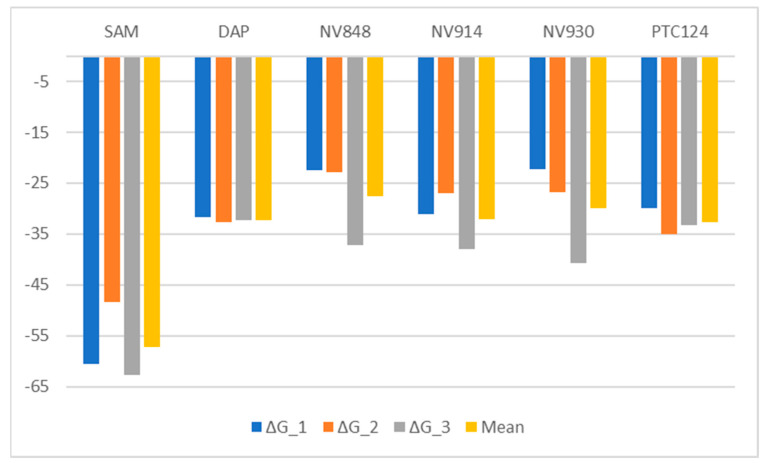
ΔG values calculated by means of MM-GBSA approach. Values are expressed in kcal/mol.

**Figure 10 ijms-24-09609-f010:**
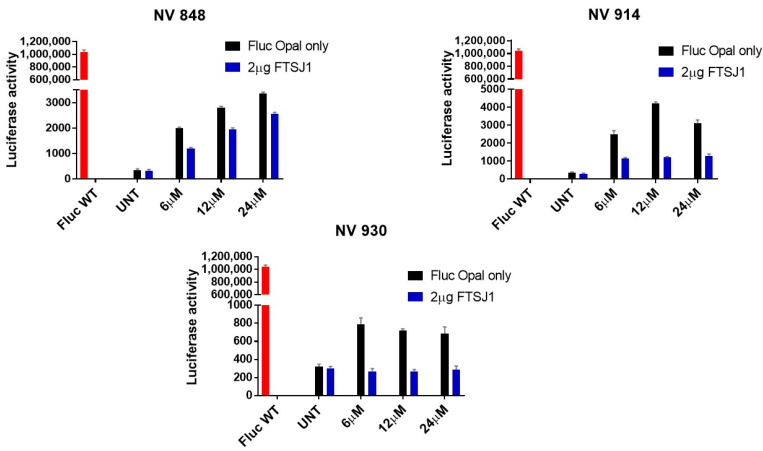
FLuc assay in HeLa cells, treated for 24 h with the indicated concentrations of NV compounds, either alone (Fluc Opal only) or in combination with FTSJ1 plasmid (2 µg FTSJ1). Note that the presence of FTSJ1 reduces luciferase activity. FLuc WT plasmid is used as a positive control.

**Figure 11 ijms-24-09609-f011:**
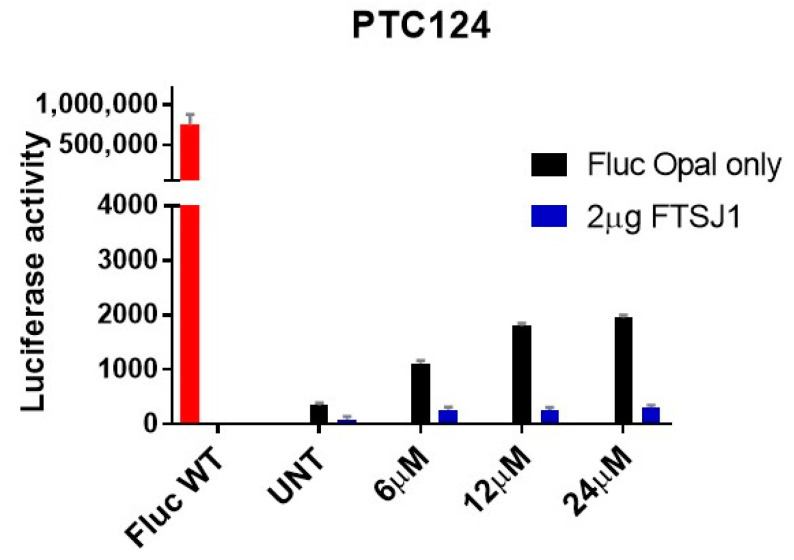
FLuc assay in HeLa cells, treated for 24 h with the indicated concentrations of PTC124, either alone (Fluc Opal only) or in combination with FTSJ1 plasmid (2 µg FTSJ1). Note that the presence of FTSJ1 reduces luciferase activity. FLuc WT plasmid is used as a positive control.

**Figure 12 ijms-24-09609-f012:**
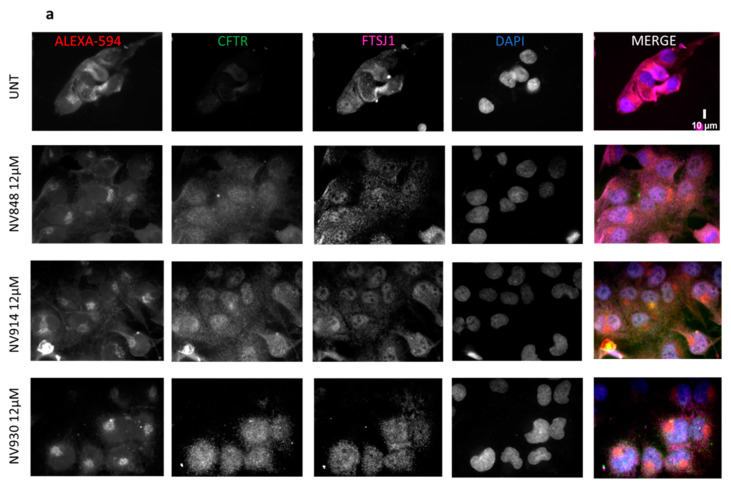
Immunofluorescence on IB3.1 cells, treated for 24 h with the indicated TRIDs, either alone (**a**,**b**) or in combination with FTSJ1 (**c**) (2 µg FTSJ1). FTSJ1 reduces the CFTR signal. Cell membrane and Golgi apparatus are stained with Alexa-594. Scale bar is 10 µm.

**Figure 13 ijms-24-09609-f013:**
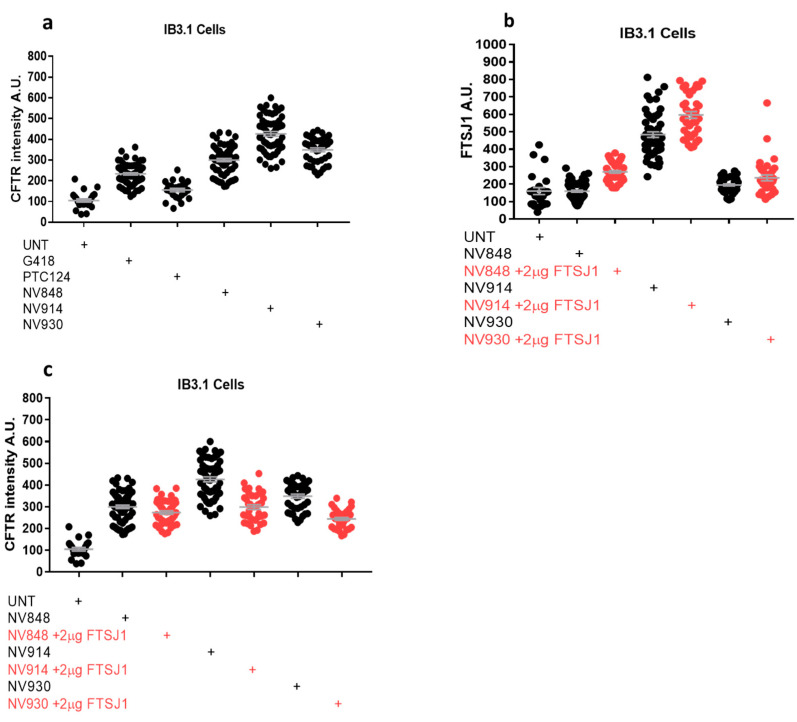
Quantification of immunofluorescence on IB3.1 cells in Figure 12. Graphs (**a**,**c**) display CFTR fluorescence intensity, whereas graph in (**b**) displays FTSJ1 fluorescence intensity A.U.: arbitrary unit.

**Table 1 ijms-24-09609-t001:** Target–ligands interactions were observed in the docking analysis and in the MD simulations. Cpd * is for compounds, Int. ** is for interactions, HB is for H-bonds, vdW is for van der Waals interactions, Hphobic is for hydrophobic interactions, and WB is for water bridges.

Docking			MD					
			Replica 1		Replica 2		Replica 3	
Cpd *	Residue	Int. **	Residue	Int. **	Residue	Int. **	Residue	Int. **
DAP	Trp55	Pi–pi	Asp116	HB	Asp116	HB	Asp116	HB
	Lys156	Pi–cation	Trp55	HPhob	Trp55	HPhob	Trp55	HPhob
	Asp116	vdW	Asp47	HB	Asp47	HB	Asp47	HB
	Asp47	vdW	Leu48	HPhob	Leu48	HPhob	Leu48	HPhob
					Gly53	HB	Gly53	HB
							Ser54	HB

NV848	Trp55	HB, pi–cation	Trp55	HB, HPhob	Asp47	WB, HPhob	Lys28	HB, ionic
	Gly53	vdW	Gly53	HB	Leu48	HB, HPhob, and Ionic	Gly53	HB
	Ser54	vdW	Ser54	HB	Gly53	HB	Ser54	HB
	Asp116	vdW	Asp116	WB	Ser54	HB, Hphobic	Trp55	HB, Hphob
					Cys115	WB, HB	Asp116	HB, WB
					Asp116	WB, HB	Lys156	Hphob

NV914	Asp75	vdW	Cys49	HB, Hphob	Cys49	HB, Hphob	Cys49	WB, HB
	Asp116	vdW	Leu76	HB, HPhob	Leu76	HPhob	Asp91	WB
	Lys156	vdW	Asp91	WB	Ile92	HPhob, WB	Ala118	HB, WB
	Ala118	vdW, HB	Ala118	HB, WB	Ala118	HB, WB	Asp120	HB, WB
	Cys49	vdW						

NV930	Asp75	vdW	Leu48	HPhob	Asp91	HB, Hphobic. WB	Ile92	Hphobic, WB
	Ala118	HB, vdW	Cys49	HB, HPhob	Ile92	WB	Ala118	HB, Hphobic, and WB
	Lys156	vdW	Val74	HPhob	Tyr130	Hphobic	Leu135	Hphobic
	Asp116	vdW	Asp75	WB	Gln134	HB, WB	Ala139	
	Cys49	vdW	Ile92	HB, HPhob	Leu135	Hphobic		
	Leu135	vdW	Leu135	HPhob	Tyr218	Hphobic, WB		
			Ile142	HPhob				

PTC124	Ser25	HB, vdW	Arg24	HB, WB	Arg24	HB, WB	Ser25	HB, WB
	Arg24	Salt bridge	Ser25	HB, WB	Ser25	HB, WB	Arg24	HB, WB
	Lys28	Salt bridge	Pro52	Hphobic	Lys28	HB, WB, Ionic, and Hphobic	Lys28	HB, WB, and Hphobic
	Lys156	Pi-cation	Trp55	Hphobic	Trp55	Hphobic	Trp55	Hphobic
	Cys49	vdW	Ala118	Hphobic	Lys156	HB, Hphobic, ionic, and WB	Lys156	Hphobic, ionic, and WB
	Pro52	vdW			Arg186	HB, ionic, and WB	Arg186	HB, ionic, and WB
	Gly53	vdW						
	Trp55	vdW						
	Arg186	Salt bridge						

## Data Availability

The views and opinions expressed are those of the authors only and do not necessarily reflect those of the European Union or the European Commission. Neither the European Union nor the European Commission can be held responsible for them.

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
