# Peer review of "Investigating the Inhibition of FTSJ1, a Tryptophan tRNA-Specific 2′-O-Methyltransferase by NV TRIDs, as a Mechanism of Readthrough in Nonsense Mutated CFTR"

_ijms, 2023, doi:10.3390/ijms24119609_

Round 1

Author Response

Reply to: 

Reviewer 1 

The current study, “Investigating the inhibition of FTSJ1 a tryptophan tRNA specific 2’-O-methyltransferase by NV TRIDs, as a mechanism of read-through in nonsense mutated CFTR", focuses on the molecular mechanism of actions of our read-through compounds NV848, NV914, and NV930 could be attributable to the inhibition of the tRNA specific 2′-Omethyltransferase (FTSJ1).  

An investigation by combined computational approaches followed by Luciferase assay evaluation in HeLa cells as well as CFTR protein cellular localization in IB3.1 cystic fibrosis cells upon transfecting an FTSJ1 cDNA harboring plasmid. However, some issues need to be addressed by emphasizing the following points before this work may be accepted. 

Q1:   Introduction: There are some aspects that require careful consideration: 

  1. Try to include some information related to docking and reduce the number of paragraphs by combining the preexisting information.  
  2. CFTR protein?  

A1: 

  1. We thank the reviewer for the comments. We added some information related to docking in the main text and reduced the number of paragraphs

  1. The “CFTR protein” in two parts of the text was a mistake, we corrected this definition with CFTR mRNA (“results” paragraph, p14, lane 354). In order to avoid misleading information, we also added the following (“results” paragraph, p14, lane 355/356) phrase: ”IB3.1 cells are characterized…instead of a UGG tryptophan codon at aminoacidic position…” Also, we added the following part in the “introduction” paragraph (p3, lanes 94-98)”: In addition, since our NV848, NV914, and NV930 compounds have been shown to rescue both protein levels and functionality of the CFTR (Cystic Fibrosis Transmembrane Conductance Regulator) chloride channel [14], mutated in CF, we also tested for CFTR protein cellular localization in IB3.1 cystic fibrosis cells [18]​​ upon FTSJ1 transfection.”

Q2:   The presentation of the results was somewhat mixed in detail, making it challenging for the reader to interpret them when combined. It is recommended to simplify the results data to enhance their interpretability by explaining some more points related to figures.  

A2: As suggested by the Reviewer we have simplified the results data (p13, lanes 326/327; p13, lanes 330/331; p14, lanes 352/353; p14 lanes 373/374; p15, lanes 376/380). 

Q3:  The text needs extensive proofreading in English and spelling errors should be taken care of. For eg. Line 110-111, The imidazole ring in histidine (His), and predicting protonation states of histidine, His. 

A3: We thank the Reviewer for the observation and provided to revise all the text by a native speaker. 

Q4: The material and methods have been covered in great length, although some points need to be taken care of.  

  1. Please include the manufacturer’s name for DMEM, FBS, DMSO, DPBS, etc.

       We have included the manufacturer’s name as suggested. 

  1. Please provide the full form before giving the acronym for e.g., QMEAN score, DMEM, FBS, DMSO, DPBS, etc. 

We corrected it accordingly to the suggestions. 

Q5: Certain references cited in this research are outdated and may require updating. Reviewing and updating the references to ensure their relevance to the current study is recommended.  

A5: We thank the reviewer for noting this. We have tried to add more up-to-date references, such as, for instance, the newest research article by the Lejeune team in which the authors provided interesting results of DAP on Cystic Fibrosis models (Leroy et al., 2023 https://doi.org/10.1016/j.ymthe.2023.01.014 ). We have also added the following sentence (“conclusion” paragraph, p20, lanes 497-500):” In addition, our results on IB3.1 cells reinforce what has been recently reported (Leroy et al., 2023) concerning patient-derived intestinal organoids (PDIOs) harboring a W1282X mutation that displayed increased swelling activity following DAP treatment.  

Q6: Please correct the legend of the tables and figures provided. For e.g. in Fig. 1 In the legend add what green and blue color denotes. 

A6: As requested, we have now corrected the legends for tables and figures. In addition, as requested by Reviewer 3, we have also added an introduction figure (now Figure 1, p2, lanes 72-74, in the “Introduction” paragraph) depicting the chemical structures of our new TRIDs (from Pibiri et al., 2020). 

Reviewer 2 Report

Investigating the inhibition of FTSJ1 a tryptophan tRNA-specific 2’- O-methyltransferase by NV TRIDs, as a mechanism of readthrough in nonsense mutated CFTR

Dear author and editor:

The authors studied the possible mechanism of action of two TRIDs (NV848 and NV914) in read-through activity and demonstrated that NV848 could lower inhibit FTSJ1. In mutated CFTR.

The article is well written and the results presentation is fine.

The article could be published in international journal of molecular science

Thank you, best regards

Author Response

Reviewer 2 

Comments: The authors studied the possible mechanism of action of two TRIDs (NV848 and NV914) in read-through activity and demonstrated that NV848 could lower inhibit FTSJ1. In mutated CFTR. 

The article is well-written and the presentation of the results is fine. 

The article could be published in the international journal of molecular science. 

Our comment: We thank Reviewer 2 for his encouraging opinion on the work. 

Reviewer 3 Report

The study by Carollo et al. characterizes several compounds thought to act as inhibitors of TSJ1. Overall the study is adequate, however there are some written grammar errors and the manuscript should be proofread for English errors. There several points that should be addressed to improve the study.

1) An introduction figure showing structures for compounds would be helplful.

2) Three simulations were run for 100 ns, is this sufficient time to complete the simulation? How was this time limit determined? This should be better explained.

3) The error bars are missing from Fig. 9, 10. This makes the data difficult to interpret. Also text too small in the figures.

4) Entropy change ‒TΔS was not calculated. A more detailed explanation of how delta G is derived should be discussed. Is it appropriate to plot delta G given that entropy change was not calculated? Also the Y axis in FIg. 8 should be labeled.

5) NV848 may act through a different mechanism than DAP and NV930. Other possible mechanisms should be discussed.

6) The data indicate that FTSJ1 overexpression partially reduces readthrough by  NV930 (Fig. 12). If NV930 is acting through inhibition of FTSJ1, then increasing NV930 concentration should inhibit FTSJ1 overexpression and reduce fluorescence in IB3 cells. Have the authors tested higher concentrations of NV930 above 2uM?

There are some written grammar errors and the manuscript should be proofread for English errors.

Author Response

Reviewer 3 

The study by Carollo et al. characterizes several compounds thought to act as inhibitors of TSJ1. Overall the study is adequate, however, there are some written grammar errors and the manuscript should be proofread for English errors. There are several points that should be addressed to improve the study. 

Q1: An introduction figure showing structures for compounds would be helpful. 

A1: We thank this for raising this point. We have now added the required figure (from Pibiri et al., 2020) in the “introduction” paragraph (now Figure 1, p2, lanes 72-74).  

Q2:Three simulations were run for 100 ns, is this sufficient time to complete the simulation? How was this time limit determined? This should be better explained. 

A2: We thank the reviewer for the comment and tried to better explain this aspect in the main text. Moreover, in Figure 6 it is possible to note that all the simulations are stable in the period of time evaluated and a total of 300ns of simulations for each system could be considered exhaustive. 

Q3: The error bars are missing from Fig. 9, 10. This makes the data difficult to interpret. Also, the text is too small in the figures. 

A3: We thank this reviewer for raising this concern. We have now added the error bars in the indicated figure (now Fig 10 and 11), where we also have enlarged the font. 

Q4: Entropy change ‒TΔS was not calculated. A more detailed explanation of how delta G is derived should be discussed. Is it appropriate to plot delta G given that entropy change was not calculated? Also, the Y axis in FIg. 8 should be labeled. 

A4: We thank the reviewer for the comment. As explained in the materials and methods section the entropy terms were not calculated  to reduce the computational time as also reported in the literature:  

Hou, T.; Yu, R. Molecular dynamics and free energy studies on the wild-type and double mutant HIV-1 protease complexed with amprenavir and two amprenavir-related inhibitors: Mechanism for binding and drug resistance. J. Med. Chem. 2007, 50, 1177–1188 

Khan, S.A.; Zia, K.; Ashraf, S.; Uddin, R.; Ul-Haq, Z. Identification of chymotrypsin-like protease inhibitors of SARS-CoV-2 via integrated computational approach. J. Biomol. Struct. Dyn. 2020, 1–10. 

In Figure 9 (previously figure 8) is already reported that the Y axis is expressed in kcal 

Q5: NV848 may act through a different mechanism than DAP and NV930. Other possible mechanisms should be discussed. 

A5: We thank this reviewer for raising this point. We have now addressed this issue as follows (“conclusion” paragraph, p20, lanes 504-512): “For instance, as demonstrated for other PTC124 derivatives by our group in the past (Pibiri et al., 2015; Pibiri et al., 2016), NV848 readthrough activity could be the result of the interaction of this molecule with the PTC in the mRNA sequence which, thus, would favor either the insertion of a near cognate amino acid or the skipping of the nonsense mutation by the ribosome without the addition of any amino acid (Campofelice et al., 2019). Other possibilities could be the suppression of the NMD pathway as well as the interaction with the translation release factors (Campofelice et al., 2019). However, more experiments are needed to exactly unravel the precise mechanism(s) of action not only for NV848 but also for all our new TRIDs in general”. 

Q6 The data indicate that FTSJ1 overexpression partially reduces readthrough by NV930 (Fig. 12). If NV930 is acting through inhibition of FTSJ1, then increasing NV930 concentration should inhibit FTSJ1 overexpression and reduce fluorescence in IB3 cells. Have the authors tested higher concentrations of NV930 above 2uM? 

A6: We thank this reviewer for raising this point. We think that this reviewer refers to the 12uM concentration.  We agree on the fact that higher NV930 concentrations could result in the inhibition of FTSJ1 activity which might be also due to a reduction of FTSJ1 protein levels, even in the case of FTSJ1 overexpression. However, we have not tried higher NV930 concentrations since in Pibiri et al. 2020 NV930 has been shown to induce cell death (HeLa cells) at higher concentrations and time points. In addition, the best readthrough effect (for both luminescence assay as well as CFTR protein levels) has been shown with 12uM concentration not only for NV930 (see Pibiri et al., 2020) but also for the other TRIDs we used in the past (see Pibiri et al., 2015; Pibiri et al., 2016, Tutone et al., 2020).   

Round 2

Reviewer 1 Report

None

Reviewer 3 Report

The authors have addressed all concerns raised.